# RoboMonster: Compositional Generalization of Heterogeneous Embodied Agents

## Abstract

Despite rapid progress in robot hardware and algorithms, a persistent gap remains between flexible decision-making in simulation and the embodiment constraints of real robots, often leading to suboptimal execution on deceptively simple tasks. We posit that, rather than emulating human morphology, robots should *compose* heterogeneous embodied agents whose capabilities extend beyond human-like end effectors. We introduce *RoboMonster*, a paradigm and system that reasons over and coordinates multiple, diverse agents to execute tasks more effectively. At the planning level, *RoboMonster* uses a multimodal large language model to perform chain-of-thought selection over a Robot Manual describing each agent's skills and limits; a Planner proposes a composition and a Verifier checks feasibility and efficiency. We benchmark this process with *RoboMonster-P* for robot-selection tasks. At the execution level, we implement interaction logic for four end-effector types in the ManiSkill environment, collect data, train downstream policies, and evaluate on *RoboMonster-E*. Experiments and ablations show that heterogeneous compositions exhibit strong compositional generalization and successfully solve tasks that defeat single-agent or single-effector baselines, including cases requiring precision or cooperative manipulation. These results suggest that capability-driven composition is a viable route to closing the embodiment gap and scaling robotic competence.

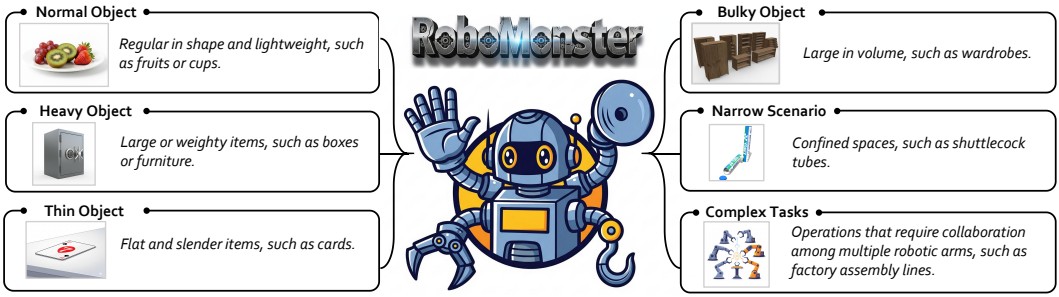

Figure 1: Current general-purpose hardware structures (e.g., grippers) may only be able to handle tasks involving interaction with conventional objects in specific scenarios (e.g., a gripper may fail to pick up a card lying flat on a table). We introduce *RoboMonster*, a novel paradigm for robotics that combines heterogeneous embodied agents to bridge the gap between hardware and algorithms through compositional generalization.

## 1 Introduction

The rapid advancement of robotics has been fueled by significant progress in both hardware and algorithmic capabilities. On the hardware front, the development of diverse robotic embodiments, ranging from humanoid robots with dexterous hands to quadrupedal robots, has paved the way for versatile robotic systems. Simultaneously, algorithmic advancements—such as open-loop models Fang et al. (2023) and closed-loop frameworks Chi et al. (2023); Zhao et al. (2023); Brohan et al. (2023); Black et al. (2024); Kim et al. —have led to substantial improvements in robot perception, decision-making,

and action execution. These strides have allowed robots to perform increasingly complex tasks with greater autonomy and precision, pushing the boundaries of what was once considered possible.

Despite these advancements, the gap between decision-making in virtual environments and the real world remains a significant challenge in embodied intelligence. In virtual simulations, execution interfaces are highly flexible, enabling quick adaptation to various scenarios. However, in the real world, decision-making is constrained by the physical properties of the robot and its pre-configured embodiment. This discrepancy often prevents algorithms from fully exploiting hardware capabilities, and vice versa, leading to suboptimal performance. A clear example of this is the task of picking up a simple card from a flat surface: while humans can achieve this with subtle skill, robotic grippers and claws, even with tactile sensors, often fail to execute the task effectively, highlighting the mismatch between algorithmic potential and hardware limitations.

One potential solution to this issue is the continuous upgrading of robotic hardware. However, this approach comes with significant costs in terms of design, manufacturing, and data collection, as new hardware requires retraining strategy models and possibly iterating algorithms to adapt to the new setup. Another direction, proposed by some researchers, is the deployment of dual-arm systems or multi-arm robots to handle tasks that a single robotic arm cannot accomplish. While this method improves task performance, it still faces limitations. For instance, even with two or more robotic arms, tasks such as picking up a card from a table remain challenging, as coordination and precision are still lacking.

In light of these challenges, we pose a fundamental question: **Do we need to design robots to resemble humans, or can we create robotic systems with capabilities that extend beyond human limitations?** For example, end-effectors such as suction cups could effectively lift cards with relatively simple mechanical structures, while multi-arm systems could lift heavy objects that would be impossible for two arms alone. Based on this concept, we introduce *RoboMonster*—a novel robotic system paradigm that combines heterogeneous embodied agents. This system enables the robot to reason and select the optimal combination of embodied agents based on visual inputs, task instructions, and the properties of its own embodied agents. Additionally, it can plan the sub-tasks for multiple agents to collaborate, enabling the system to generalize to new or more difficult tasks.

To validate *RoboMonster*, we constructed a heterogeneous multi-agent system that leverages multi-modal large language models (MLLM) for high-level planning and employs four specially designed end-effectors to perform diverse tasks. At the high-level planning stage, we present a system for planning with compositional heterogeneous embodied agents, leveraging the *RoboMonster-P* benchmark for robot selection tasks. The system selects agents based on task requirements and agent capabilities, using a Robot Manual that outlines each agent's skills and limitations. The planning framework consists of a Planner that performs chain-of-thought reasoning to choose agents, and a Verifier that ensures task feasibility and efficiency.

At the execution level, we modeled the interactions between four types of end-effectors within the ManiSkill Gu et al. (2023) simulation environment. We then collected data, trained downstream policies, and tested our system through various tasks. This approach allows us to validate the compositional generalization of heterogeneous agents in real-world scenarios, demonstrating that such systems can outperform single-gripper arms in solving tasks that require coordination among different agents. Through this validation, we highlight the superior execution capabilities of heterogeneous end-effectors, as seen in tasks where multi-agent collaboration is essential for success.

Our main contributions are as follows:

- *Concept & Paradigm.* We introduce *RoboMonster*, a novel paradigm for robotics that combines heterogeneous embodied agents to bridge the gap between hardware and algorithms through compositional generalization.

- *Planning Verification.* We propose a simple and efficient MAS planning system for selecting heterogeneous embodied agents based on task requirements and capabilities, and demonstrate its feasibility and efficiency using the *RoboMonster-P* benchmark.

- *Execution Verification.* We implement interaction logic for four types of end-effectors, construct *RoboMonster-E* benchmark, collect data, and train corresponding policy models to demonstrate the execution advantages of heterogeneous end-effectors.

- *Experimental Results.* Extensive experiments and ablation studies demonstrate that *RoboMonster* can efficiently schedule and execute tasks that a single embodied agent or single end-effector cannot accomplish, both at the planning and execution levels.

## 2 RELATED WORK

### 2.1 EMBODIED MULTI-AGENT COOPERATION

Real-world embodied environments often demand collaboration among heterogeneous robots. Prior studies have investigated this challenge through task allocation Obata et al. (2024); Wang et al. (2024b); Liu et al. (2025) and high-level multi-agent decision making Zhang et al. (2023); Wang et al. (2025a). More recently, large language models (LLMs) have been introduced to enhance multi-agent coordination, showing notable progress in distributed planning and communication Bo et al. (2024a); Guo et al. (2024b); Nasiriany et al. (2024); Zhou et al. (2023). Vision-language models (VLMs) have begun to extend these capabilities to embodied multi-agent contexts Wang et al. (2025b); Zhang et al. (2024), but they generally assume homogeneous capabilities or treat each agent independently, without mechanisms for integrating complementary skills across heterogeneous agent. VIKI Kang et al. (2025) explicitly consider heterogeneous robots, but their focus remains at the high-level planning stage without addressing the deployment of fine-grained low-level control strategies. In contrast, our work focuses on enabling collaborative control among diverse embodiments, demonstrating how heterogeneous end-effectors can be jointly orchestrated to accomplish complex tasks that exceed the ability of single agent type.

### 2.2 ROBOT LEARNING IN MANIPULATION

Task-specific policy architectures Chi et al. (2023); Ke et al. (2024); Liang et al. (2023; 2024; 2025); Wang et al. (2024a); Wen et al. (2025); Ze et al. (2024) often achieve strong results in controlled settings, but their designs are tightly coupled to particular tasks or morphologies, which makes transferring them to new embodiments difficult. In contrast, large-scale foundation models trained on diverse multi-robot datasets—such as RT-1 Brohan et al. (2022) for real-time manipulation, RT-2 Brohan et al. (2023) for semantic planning, and diffusion-based models like RDT-1B Liu et al. (2024) and $\pi$ Black et al. (2024)—show more promising generalization across tasks. Building on this trend, vision–language–action systems including OpenVLA Kim et al., CogACT [26], Octo Octo Model Team et al. (2024), LAPA Ye et al., and OpenVLA-OFT Kim et al. (2025) highlight how pretrained representations can be efficiently adapted to different robots and sensing modalities. However, these approaches typically presuppose homogeneous agents and uniform capabilities. Our work explicitly explores this dimension, showing how heterogeneous embodiments can be organized into a coherent control framework that leverages their complementary skills to solve more complex tasks.

### 2.3 MULTI-AGENT SYSTEM FOR ROBOT PLANNING

Large language model based multi-agent systems (MAS) provide general infrastructures for role-specialized collaboration and tool use Li et al. (2023); Wu et al. (2023); Chen et al. (2023); Hong et al. (2024); Qian et al. (2024). Building on these infrastructures, a growing line of work treats *planning* itself as a multi-agent process—either by decomposing tasks into sub-plans, coordinating expert agents, or reflecting over intermediate results to improve plan quality Guo et al. (2024a); Wei et al. (2025); Li et al. (2025); Tao et al. (2024b); Bo et al. (2024b). Closer to robotics, recent efforts couple MAS with embodied planning and execution: RoCo coordinates multi-robot dialogue for sub-tasking and motion-waypoint generation Mandi et al. (2024), MALMM Singh et al. distributes high-level planning and low-level control across specialized agents with feedback-driven re-planning, and SMART-LLM SMART Lab (2023) converts high-level instructions into multi-robot task plans. In our work, we instantiate an MAS specifically to plan for heterogeneous robots with diverse end-effectors, enabling coordinated high-level assignment across embodiments.

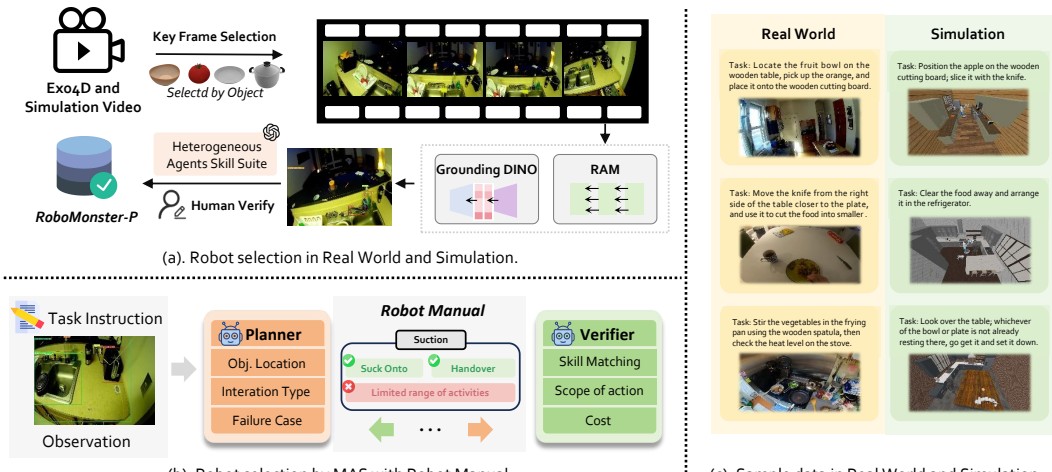

Figure 2: **Planning for Compositional Heterogeneous Embodied Agents.** (a) Data collection and curation process of *RoboMonster-P*. (b) Construction of the multi-agent system, which analyzes input images and instructions to select appropriate heterogeneous embodied agents. (c) *RoboMonster-P* includes a diverse set of real-world and simulated environments and tasks.

# 3 SPECTRUM OF REAL-WORLD ROBOTIC TASKS

Real-world tasks span a broad and diverse distribution, while only a small portion can be addressed by current robotic hardware and mechanical structures. We categorize tasks according to the properties of their interactive objects and environments, aiming to leverage compositional generalization to build heterogeneous embodied-agent systems capable of handling a wider range of tasks.

**Normal Objects:** These objects have regular shapes, moderate weight, and standard volume. Tasks involving them can be effectively addressed by training policy models for grippers or dexterous hands. *e.g., picking up fruits, cups.*

**Heavy Objects:** These items exceed the payload limits of conventional manipulators or grippers and may require the collaboration of multiple robotic arms. *e.g., transporting a safe.*

**Thin Objects:** Characterized by very small thickness or volume, these objects demand higher precision than current grippers or dexterous hands can provide. Specialized end-effectors are necessary. *e.g., picking up a playing card from a table.*

**Bulky Objects:** Large in volume, these objects cannot be stably manipulated by a single arm alone, requiring multiple embodied agents for safe interaction. *e.g., moving a wardrobe.*

**Narrow Scenarios:** Confined spaces where grippers, dexterous hands, or even human hands cannot pass through, necessitating special end-effectors. *e.g., placing a shuttlecock into a shuttlecock tube.*

**Complex Tasks:** Tasks that are inherently difficult or require high efficiency, often demanding collaboration among multiple heterogeneous embodied agents. *e.g., factory production line.*

Designing universal end-effectors (*e.g.*, dexterous hands) entails continuous iterations of mechanical structures. Instead, we explore compositional generalization through heterogeneous embodied agents, aiming to build an embodied system that can cover the entire distribution of real-world tasks.

# 4 ROBOMONSTER

Compositional generalization in heterogeneous multi-agent systems can be achieved in two distinct stages: (1) a high-level planning phase that selects the appropriate agents, and (2) a low-level control phase where the selected embodied agents execute the task using their individual policies. Designing heterogeneous embodied agents typically involves defining interaction logic in either simulations or real-world setups. In contrast, the planning phase is primarily focused on selecting from a predefined

set of heterogeneous agents based on the task requirements. Therefore, we decouple the high-level planning from the low-level execution to facilitate effective validation.

In the high-level phase (Sec. 4.1), we develop an embodied planning system that schedules the appropriate heterogeneous agents to enable compositional generalization, allowing the completion of tasks that homogeneous agents alone cannot solve. In the low-level phase (Sec. 4.2), we instantiate robotic arms with four distinct end-effectors within the ManiSkill Tao et al. (2024a) environment, collect demonstrations, and train the corresponding policies. This stage validates that heterogeneous agents, during execution, can leverage compositional generalization to solve tasks that a single-gripper arm would not be capable of completing.

## 4.1 PLANNING FOR COMPOSITIONAL HETEROGENEOUS EMBODIED AGENTS

**Data Collection and Curation Process.** We reformulate the robot selection task as a visual reasoning problem, as illustrated in Fig. 2(a), where the task allocator selects a set of robots from a predefined set of embodied agents $\mathcal{A}_{\text{embodied}}$, considering task requirements and agent capabilities. Each instance consists of a keyframe observation $O$, selected from real-world and simulation data, and a task instruction $I$, generated based on the objects present in $O$. The output is a set of selected robots $R = \{r_j\}, j \in [1, M]$.

Ground truth labels are created using task-specific templates that specify which robot types are necessary or unnecessary, based on the task goal and contextual factors, and grounded in embodiment rules. For reasoning, we employ a chain-of-thought approach, where the model first analyzes task requirements, identifies available robots from the embodied agent set, evaluates their suitability, and then selects the appropriate robots. The task allocator $g_{\text{act}}$, powered by GPT-4o OpenAI et al. (2024), generates the robot selection $R = g_{\text{act}}(I, O)$. A verification module $C_{\text{act}}$ ensures that the generated labels adhere to task constraints, with human oversight for error correction and label quality assurance. The dataset we construct for robot selection task is called *RoboMonster-P*.

**Brain of *RoboMonster*.** We build a multi-agent decision system that selects one or more embodied agents based on the task instruction and the current observation. The objective is to demonstrate that heterogeneous multi-agent collaboration can yield compositional generalization, and that this effect is particularly effective for high-level embodied planning. To ensure the validity of this conclusion, we deliberately avoid introducing complex designs into the decision system. Instead, the overall MAS framework is constructed by following the principles of ReAct Yao et al. (2023) and Reflection Shinn et al. (2023).

Before decision-making, we compile a *Robot Manual* from the URDF and parameter files of all available embodied agents. The manual specifies, for each type of agent, its skills, action range, and execution cost. This serves as the knowledge base for reasoning about heterogeneous capabilities.

As shown in Fig. 2(b), the first component, the MLLM-based **Planner**, performs chain-of-thought reasoning to summarize object locations, interaction logic, and potential failure cases (e.g., spatial constraints). Based on this reasoning, it selects the candidate embodied agents required for the task. The second component, the LLM-based **Verifier**, validates these selections against the Robot Manual, checking multiple aspects to ensure that the chosen agents can accomplish the task with minimal cost.

We validate our system on the *RoboMonster-P* benchmark and show that, even without a carefully engineered decision pipeline or domain-specific fine-tuning, heterogeneous multi-agent systems are still able to generalize compositionally across diverse tasks.

## 4.2 EXECUTION WITH COMPOSITIONAL HETEROGENEOUS AGENTS

To verify that heterogeneous embodied agents can achieve scene- and task-level generalization through compositionality during execution, we build a set of heterogeneous agents and a broad distribution of manipulation tasks on top of the ManiSkill Gu et al. (2023) simulation platform. Using an automated MLLM-based data-collection pipeline, we gather training data and then train and evaluate a heterogeneous multi-agent system based on imitation learning.

**Heterogeneous Effector Embodied Agents.** We first modify both the control logic and visual appearance of the robot end-effectors to implement four types of heterogeneous grippers, as illustrated in Fig. 3(a). The logic of the four end-effectors is as follows:

(a). Heterogeneous End–Effector Instantiation.

(b). Heterogeneous Multi–Agent Imitation Learning Policy.

Figure 3: **Execution with Compositional Heterogeneous Agents.** (a) Instantiation of diverse end-effectors in ManiSkill, each designed with unique structures that provide distinct capabilities. (b) Construction of heterogeneous multi-agent imitation learning policies, enabling collaboration among different embodied agents to accomplish complex tasks.

1. *[Gripper]* General and precise gripping works well for most rigid or graspable objects and shows some tolerance to size variation. However, it tends to be unstable for objects that are excessively thin or very slippery.

2. *[Stick]* Applicable for pushing, inserting or clearing in narrow cavities or channels (*e.g.*. pushing a shuttlecock into a bucket, unblocking a small tube); unsuited for transporting or grasping.

3. *[Suction]* Suitable for objects with smooth, relatively flat surfaces (*e.g.*, thin cards on a table, smooth spheres or cubes), and most stable when there is good sealing with the contact surface. It is not suitable or performs poorly for porous or rough surfaces, high curvature, heavy objects, or when there are insufficient contact or sealed points.

4. *[Ring-shaped Gripper]* This end-effector is an annular gripper whose inner diameter can be freely varied to accommodate objects of different sizes. Caging-style constraint provides stable support for large round or cylindrical objects, such as vases or smooth sphere, which are difficult to grasp securely with fingered grippers or suction-based methods. By surrounding the object, the executor forms a boundary that prevents slipping or rolling and thus improves stability.

Details of these modifications can be found in the supplementary material (Section. D).

**Data Collection and Curation Process.** Next, we design an automated data-collection pipeline powered by a multimodal large language model (MLLM). The pipeline collects trajectory data for all heterogeneous agents within each task category, enabling *low-level policy* training. Inspired by RoboFactory , the pipeline consists of two components: *RoboBrain*, which decomposes tasks and schedules primitive functions; and *RoboChecker*, which verifies whether the generated trajectories are reasonable and free of anomalies. When scheduling primitive functions, we additionally include the end-effector type as an input variable so that each heterogeneous gripper can produce its own unique action sequence.

Based on the above methodology, we introduce the *RoboMonster-E* benchmark, built on the ManiSkill simulator. *RoboMonster-E* aims to instantiate manipulation tasks with diverse distributions, detailed described in Sec. 5.2. It includes **5 tasks** across environments with varying numbers of agents, constructed around the Franka Emika Panda arm—a 7-DoF robotic manipulator equipped with interchangeable end-effectors that enable flexible manipulation.

**Compositional Agents Trajectory Execution.** Finally, we adapt the single-agent imitation-learning framework to a multi-agent system, where each agent learns an independent policy from its own egocentric view. We employ a *Global-View + Shared-Policy* paradigm (Fig. 3(b): all agents share the same global observation and use a single shared policy to generate a joint action sequence, which is then assigned to the corresponding agents. Compared with approaches that use only a single gripper, employing an optimal combination of end-effectors yields better generalization and performance. Notably, since *RoboMonster-E* is designed to validate the execution-level advantages of heterogeneous embodied agents, we adopt the optimal end-effector combination by default.

# 5 EXPERIMENTS

## 5.1 PLANNING WITH COMPOSITIONAL HETEROGENEOUS EMBODIED AGENTS

**Experiments Setting.** We cast planning as *agent selection* from a small library of heterogeneous end-effectors. Given a single RGB scene image and a natural-language instruction, the model must output exactly one or multiple embodied agents from the label set which is described in detail in the Sup. C.2. We evaluate on a 200-example test set *sampled from RoboMonster-P* (More detial about sampling protocol and distribution is in Sup. C). We apply the same instruction-following template that (i) lists $\mathcal{E}$, (ii) asks for a *single* choice, and (iii) forbids extraneous text. We canonicalize predictions to the four labels via simple string matching and report **top-1 accuracy** over these 200 items.

**Baselines.** We compare against strong open- and closed-source VLMs as well as our modular agentic system: *Qwen2.5-VL-32B-Instruct* (open), *GLM-4.5V* (open), *GPT-5* (closed), *Gemini-2.5-Pro* (closed), *Claude Sonnet 4 (2025-05-14)* (closed), and **MAS (ours)**, which couples a planner (MLLM reasoning) with a verifier (LLM rule-/constraint checker) operating over a robot manual that encodes capability and feasibility constraints. For a fair comparison, all single-pass VLMs share the same prompt schema and are restricted to one-shot selection; *MAS* may internally perform at most one planner–verifier iteration but still outputs a single final label.

Table 1: **Agent selection accuracy** on *RoboMonster-P* (200 examples). No finetuning; temperature $= 0.0$.

| Model | Accuracy |
|---|---|
| Gripper Only | 0.120 |
| Qwen2.5-VL-32B-Instruct | 0.235 |
| GLM-4.5V | 0.260 |
| GPT-5 | 0.440 |
| Gemini-2.5-Pro | 0.425 |
| Claude Sonnet 4 (2025-05-14) | 0.415 |
| **MAS (Planner+Verifier, ours)** | **0.450** |

**Generalization via Agent Selection.** Tab. 1 summarizes top-1 accuracy on *RoboMonster-P*. Policies based on a traditional single gripper are limited by their mechanical structure, achieving only a 10% task completion rate, show that heterogeneous multi-agent collaboration enables compositional generalization. While closed-source models generally outperform open-source ones, our proposed *MAS* achieves the best overall accuracy. This suggests that lightweight verification against the robot manual is effective for correcting choices that appear plausible but are infeasible in practice.

## 5.2 EXECUTION WITH COMPOSITIONAL HETEROGENEOUS EMBODIED AGENTS

**Experiments Setting.** With the advancement in simulator realism, numerous outstanding simulation environment frameworks have arisen, for example, *RoboTwin* Mu et al. (2024) and *RoboFactory* Qin et al. (2025), which are built on ManiSkill Tao et al. (2024a). Taking into account usability and other relevant factors, we utilize the *RoboFactory* framework to collect expert demonstration data. In order to compare the effect of using heterogeneous end-effectors versus gripper-only under different policies and varying amounts of expert data, we collect 25, 50, and 75 trajectories for DP Chi et al. (2023). Since each trajectory under DP3 Ze et al. (2024) contains relatively less information (due to sparse point cloud sampling from raw data), we instead use 50, 100, and 150 expert demonstration trajectories in this paper.

We designed five challenging tasks involving both single-agent and dual-agent settings to validate the execution performance advantages of our specialized heterogeneous end-effectors (gripper, suction, stick, and ring-shaped gripper). For clarify, the following task descriptions are provided under the assumption that *RoboMonster* brain has already filtered out inappropriate end-effectors. Therefore, our discussion is limited to the **optimal end-effector combinations** and the gripper-only setting. The specific tasks are as follows:

1. **Suction-lift Card:** The agent $\mathcal{A}_1$ uses suction or gripper end-effector to lift the credit card placed on the cube.

2. **Pick Pokéball:** The agent $\mathcal{A}_1$ uses ring-shaped gripper or normal gripper end-effector to pick up the Pokéball (a smooth sphere).

3. **Pick Vase:** The agent $\mathcal{A}_1$ uses ring-shaped gripper or normal gripper end-effector to pick up the vase.

4. **Place Shuttlecock:** The agent $\mathcal{A}_1$ grasps the shuttlecock and positions it in the opening of the shuttlecock barrel by using gripper end-effector. The agent $\mathcal{A}_2$ then uses stick end-effector to push the shuttlecock into the barrel, or $\mathcal{A}_2$ uses the closed gripper end-effector pushes the shuttlecock in.

5. **Swipe Card:** The agent $\mathcal{A}_1$ lifts the credit card (via suction or gripper end-effector) from the cube and moves it to a position convenient for hand-off to $\mathcal{A}_2$. Then, $\mathcal{A}_2$ uses gripper to align the card with the slot in the POS terminal and insert it into the POS terminal.

**Baseline.** Imitation learning methods (DP, DP3) remain popular policies. Therefore, we used these two approaches as baselines to validate the effectiveness and performance of heterogeneous end-effectors. Specifically, there are two paradigms are considered in this work.

Table 2: Performance comparison across various paradigms.

| Paradigm | E-e. Setup | Swipe Card (*Diffusion Policy*) | | |
| --- | --- | --- | --- | --- |
| | | 25 Demo | 50 Demo | 75 Demo |
| *Global-View* | Gripper Only | 17% | 23% | 25% |
| *+ Shared-Policy* | Heterogeneous E-e. | **60%** | **67%** | **77%** |
| *Local-View* | Gripper Only | 0% | 0% | 0% |
| *+ Separate-Policy* | Heterogeneous E-e. | 0% | 2% | 8% |

*Global-View + Shared-Policy.* All agents share the same global observation and use a single policy to produce an action sequence, which is then assigned to the corresponding agents. The observation can be presented as $\mathbf{O}_{global} = \text{concat}([\mathbf{A}_0, \mathbf{A}_1, \ldots, \mathbf{A}_N, \mathscr{E}(\mathbf{X}_{global})])$.

*Local-View + Separate-Policy.* Each agent has its own independent observation, and each agent uses its own separate policy to generate individualized action sequences, where the individual observation can be formulated as $\mathbf{O}_i = \text{concat}([\mathbf{A}_i, \mathscr{E}(\mathbf{X}_i)])$.

Where $\mathbf{A}_i$ is the joint action of the $i$-th agent, $N$ represents the number of agents, $\mathscr{E}(\cdot)$ is the encoder, $\mathbf{X}_{global}$ and $\mathbf{X}_i$ are the global view and the $i$-th agent view respectively (which is the RGB image in DP, and point cloud in DP3). In addition, we evaluated the performance of the two paradigms of DP under different end-effector setups and varying numbers of demonstrations on the **Swipe Card** task. The detailed success rates are reported in Tab. 2. The results show that the *Global-View + Shared-Policy* paradigm holds a significant advantage in complex, long-horizon, collaborative tasks. We believe this is because such tasks demand extremely strict temporal constraints, which the *Local-View + Separate-Policy* paradigm finds difficult to learn from the individual datasets. Based on above findings, we subsequently employed the *Global-View + Shared-Policy* paradigm as the training strategy for DP and DP3. More details of DP and DP3 training (e.g., hyperparameters) are reported in supplementary material (Sec. A).

Table 3: Performance of different end-effector setup. We report the success rates of heterogeneous end-effectors and gripper-only across five tasks and two policies with six demonstration settings. (Abbr.: E-e. = End-effector, R-s. = Ring-shaped, Gri. = Gripper, Sti. = Stick, Suc. = Suction)

| Task Name | E-e. Setup | *Diffusion Policy* | | | *3D Diffusion Policy* | | | **Average** |
| --- | --- | --- | --- | --- | --- | --- | --- | --- |
| | | 25 Demo | 50 Demo | 75 Demo | 50 Demo | 100 Demo | 150 Demo | |
| Suction-lift Card | Gripper Only | 7% | 14% | **15%** | 21% | 20% | **23%** | 16.7% |
| | Ours (Suction) | **100%** | **100%** | **100%** | **100%** | **100%** | 93% | 98.8% |
| Pick Pokéball | Gripper Only | 37% | **69%** | 64% | 54% | **75%** | 73% | 62% |
| | Ours (R-s. Gri.) | 78% | **100%** | **100%** | 88% | **100%** | **100%** | 94.3% |
| Pick Vase | Gripper Only | 28% | **41%** | 37% | 14% | 52% | **54%** | 37.7% |
| | Ours (R-s. Gri.) | **100%** | **100%** | **100%** | **100%** | **100%** | **100%** | 100% |
| Place Shuttlecock | Gripper Only | 46% | 43% | **48%** | 42% | 45% | **46%** | 45% |
| | Ours (Gri. & Sti.) | 37% | 51% | **54%** | 67% | **76%** | 75% | 60% |
| Swipe Card | Gripper Only | 17% | 23% | **25%** | 2% | 19% | **31%** | 19.5% |
| | Ours (Suc. & Gri.) | 60% | 67% | **77%** | 5% | 62% | **71%** | 57% |

**Compositional Generalization.** The heterogeneous multi-end effector defined in this work (which comprises four specialized end-effectors) is described in detail in Sec. 4.2. Moreover, we illustrate the workflows of three representative tasks (see Fig. 4), these tasks exemplifies the usage and distinctions among the four end-effectors.

We extensively evaluate our proposed heterogeneous multi-end effector paradigm under both DP and DP3 policies, including both single-agent and dual-agent configurations. Each policy is tested on five tasks, with three different numbers of demonstrations. As shown in Tab. 3, in the three single-agent tasks, the average success rate using the heterogeneous multi-end effector exceeds **94%**, which is a marked improvement over the gripper-only setup ($16.7\% \rightarrow 98.8\%$, $62\% \rightarrow 94.3\%$, $37.7\% \rightarrow 100\%$). In the long-horizon tasks with dual-agent, the average success rate drops substantially compared to the single-agent setting, reaching only about **60%** or **57%**. However, it still shows a clear improvement over the gripper-only configuration ($45\%$, $19.5\%$).

We also found that in simple single-agent tasks, a moderate number of demonstrations (**50 Demo** for DP, **100 Demo** for DP3) is often sufficient to achieve good performance (in fact, the best performance in the **Pick Pokéball** task occurs at these levels). However, for complex long-horizon dual-agent tasks (with the exception of the **Place Shuttlecock** task under DP3), peak performance is attained only when using large numbers of demonstrations (**75 Demo** for DP, **150 Demo** for DP3).

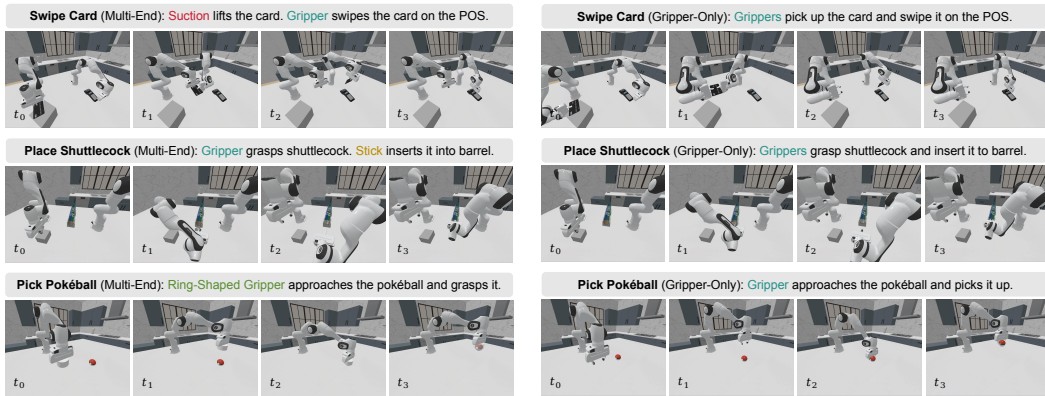

Figure 4: Demonstrations of the tasks. Three representative tasks, namely **Swipe Card**, **Place Shuttlecock**, **Pick Pokéball**, are selected to cover four different types of end-effectors. The left side illustrates the heterogeneous end-effector setup proposed in this work, while the right side presents the gripper-only counterpart for the tasks.

# 6 CONCLUSION

We introduced *RoboMonster*, a paradigm that leverages heterogeneous embodied agents to overcome the embodiment gap between simulation and real-world robots. By combining multimodal planning with a Planner–Verifier framework and executing through diverse end-effectors, *RoboMonster* enables capability-driven composition that outperforms single-agent or single-effector baselines. Experiments on *RoboMonster-P* and *RoboMonster-E* demonstrate strong compositional generalization, improved precision, and effective cooperative manipulation. These results suggest that heterogeneous composition is a scalable route to enhancing robotic competence.

**Limitation and Future Work.** At the *Execution with Compositional Heterogeneous Agents* level, we adopt a purely vision-based imitation learning scheme in simulation. Exploring additional modalities—for example, employing VLA models—to further examine compositional generalization at the execution level is a worthwhile direction. Moreover, validating these capabilities on physical robots represents another meaningful avenue. In future work, we plan to extend *RoboMonster* to more complex embodiments, richer sensory inputs, and broader real-world tasks, further advancing the pursuit of general-purpose embodied intelligence.

## ETHICS STATEMENT

We confirm that this work does not involve human subjects, personal or sensitive data, or ethical content of concern. There are no foreseeable risks to privacy, safety, or societal harm associated with our methods and results. We commit to full transparency, and will provide open-source code and documentation in accordance with the ICLR Code of Ethics.

## REPRODUCIBILITY STATEMENT

To facilitate full reproducibility, we provide:

1. We will release the complete source code of **RoboMonster** (including **RoboMonster Brain**, **RoboMonster-P**, **RoboMonster-E**, and all components used in the paper) upon acceptance of the manuscript, to support data collection, model training, and evaluation. e key code for heterogeneous end-effectors in the ManiSkill simulator has been provided in the supplementary material (Section D).

2. Detailed hyper-parameters and network architectures in supplementary material (Section A).

All experiments were carried out in open-source simulation environments, and we will release the corresponding documentation alongside the code to support researchers in reproducing our results.

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

# *RoboMonster* Supplementary Material

## USE OF LARGE LANGUAGE MODELS (LLMS)

We used large language models (LLMs) only for language polishing and minor editing of our manuscript (*e.g.* improving grammar, clarity, phrasing). All the text was reviewed, corrected, and verified by the authors. We take full responsibility for the final content of the paper, including any portions influenced by LLM assistance.

## A TRAINING DETAILS

In this section, we present the training details, covering the hyperparameter configurations of the two baselines as well as representative training times.

Table 4: Hyperparameters for *Diffusion Policy* training. (Abbr.: H-Paras = Hyperparameters, Pre. = Prediction, Obs. = observation, Act. = Action, BS = Batch Size)

| H-Paras | Suction-lift Card | Pick Pokéball | Pick Vase | Place Shuttlecock | Swipe Card |
|---|---|---|---|---|---|
| | | | *Diffusion Policy* | | |
| Pre. Horizon | 32 | 32 | 32 | 32 | 32 |
| Obs. Horizon | 20 | 20 | 20 | 20 | 20 |
| Act. Horizon | 8 | 8 | 8 | 8 | 8 |
| Image Shape | $3 \times 256 \times 256$ | $3 \times 256 \times 256$ | $3 \times 256 \times 256$ | $3 \times 256 \times 256$ | $3 \times 256 \times 256$ |
| Action Shape | 8 | 8 | 8 | 15 | 16 |
| 4060 BS | N/A | N/A | N/A | N/A | N/A |
| $2 \times$ 4090 BS | 32 | 32 | 32 | 32 | 32 |
| $2 \times$ H800 BS | 64 | 64 | 64 | 64 | 64 |
| Learning Rate | 1.0e-4 | 1.0e-4 | 1.0e-4 | 1.0e-4 | 1.0e-4 |
| Warm-up Steps | 500 | 500 | 500 | 500 | 500 |
| Betas | [0.95, 0.999] | [0.95, 0.999] | [0.95, 0.999] | [0.95, 0.999] | [0.95, 0.999] |
| Weight Decay | 1.0e-6 | 1.0e-6 | 1.0e-6 | 1.0e-6 | 1.0e-6 |
| Epsilon | 1.0e-8 | 1.0e-8 | 1.0e-8 | 1.0e-8 | 1.0e-8 |
| Epochs | 300 | 300 | 300 | 300 | 300 |

Table 5: Hyperparameters for *3D Diffusion Policy* training.

| Hyperparameters | Suction-lift Card | Pick Pokéball | Pick Vase | Place Shuttlecock | Swipe Card |
|---|---|---|---|---|---|
| | | | *3D Diffusion Policy* | | |
| Prediction Horizon | 32 | 32 | 32 | 32 | 32 |
| Observation Horizon | 20 | 20 | 20 | 20 | 20 |
| Action Horizon | 8 | 8 | 8 | 8 | 8 |
| Point Cloud Shape | $3 \times 1024$ | $3 \times 1024$ | $3 \times 1024$ | $3 \times 1024$ | $3 \times 1024$ |
| Action Shape | 8 | 8 | 8 | 15 | 16 |
| 4060 Batch Size | 32 | 32 | 32 | 32 | 32 |
| $2 \times$ 4090 Batch Size | 128 | 128 | 128 | 128 | 128 |
| $2 \times$ H800 Batch Size | 256 | 256 | 256 | 256 | 256 |
| Learning Rate | 1.0e-4 | 1.0e-4 | 1.0e-4 | 1.0e-4 | 1.0e-4 |
| Warm-Up Steps | 500 | 500 | 500 | 500 | 500 |
| Betas | [0.95, 0.999] | [0.95, 0.999] | [0.95, 0.999] | [0.95, 0.999] | [0.95, 0.999] |
| Weight Decay | 1.0e-6 | 1.0e-6 | 1.0e-6 | 1.0e-6 | 1.0e-6 |
| Epsilon | 1.0e-8 | 1.0e-8 | 1.0e-8 | 1.0e-8 | 1.0e-8 |
| Epochs | 300 | 300 | 300 | 300 | 300 |

Table 6: Task Descriptions for the *RoboMonster-E* Benchmark

| Task | Description | Target Condition |
|---|---|---|
| Suction-lift Card | A credit card is placed at the edge of a cube, and a robotic arm must choose an appropriate end-effector to grasp the card and lift it to a specified height. | The height of the credit card reaches a predefined threshold. |
| Pick Pokéball | A Pokéball is placed on the table, and a robotic arm must choose an appropriate end-effector to grasp it and lift it to a specified height. | The height of the Pokéball reaches a predefined threshold. |
| Pick Vase | A vase is placed on the table, and a robotic arm must choose an appropriate end-effector to grasp it and lift it to a specified height. | The height of the vase reaches a predefined threshold. |
| Place Shuttlecock | A shuttlecock and a barrel are placed on the table. Two robotic arms should each choose an appropriate end-effector: one to place the shuttlecock at the rim of the barrel, and the other to insert it inside. | We assume that the number of shuttlecocks already inside the barrel follows a 50% probability of being six and a 50% probability of being seven. The task requires pushing the shuttlecock into a position adjacent to the outermost shuttlecock. |
| Swipe Card | A credit card is placed on top of a cube, while a POS terminal is located on the table. Two robotic arms should each choose an appropriate end-effector: one to lift the credit card and hand it over to the other arm, and the other to insert the card into the slot of the POS terminal. | The distance between the credit card and the slot of the POS terminal is smaller than a predefined threshold. |

***Diffusion Policy.*** We employ a **CNN-based** *Diffusion Policy* and evaluate its training performance across representative resource-constrained (NVIDIA RTX 4060 GPU), moderate-resource (2 × NVIDIA RTX 4090 GPU), and high-resource (2 × NVIDIA H800 GPU) computing platforms. The corresponding hyperparameter settings are summarized in Tab. 4. On the resource-constrained platform, the training could not be completed due to the large data volume. For researchers limited to low-resource platforms, one may attempt to simultaneously reduce the Horizon, Image Shape, and Batch Size during training (though we do not recommend this as a preferred strategy).

It is worth noting that we adopted the `torch.optim.AdamW` optimizer. The hyperparameter values for the Learning Rate, Warm-up Steps, Betas, Weight Decay, and Epsilon are all specified in the corresponding Tab. 4. We report several representative training times as follows:

1. On 2 × 4090 GPU: with 75 demos and 300 epochs for the **Swipe Card** task, approximately **20 hours**.

2. On 2 × H800 GPU: with 75 demos and 300 epochs for the **Swipe Card** task, approximately **13 hours**.

3. On 2 × H800 GPU: with 75 demos and 300 epochs for the **Suction-lift Card** task, approximately **6 hours**.

***3D Diffusion Policy.*** For *3D Diffusion Policy*, we adopt an almost identical training strategy. The specific training configurations are listed in Tab. 5. A notable distinction compared to *Diffusion Policy* is that *3D Diffusion Policy* achieves much shorter training times, and is considerably more friendly for researchers with low-resource platforms. We report several representative training times as follows:

1. On 1 × 4060 GPU: with 150 demos and 300 epochs for the **Swipe Card** task, approximately **23 hours**.

2. On 2 × 4090 GPU: with 150 demos and 300 epochs for the **Swipe Card** task, approximately **7 hours**.

3. On $2 \times$ H800 GPU: with 150 demos and 300 epochs for the **Swipe Card** task, approximately **2.5 hours**.

## B    EVALUATION DETAILS

**Tasks Evaluation.** We interpolated the action sequences generated by the model to make the motion trajectories smoother. For each task, we evaluated it across 100 seeds, varying the initial object positions and environmental conditions. We introduced a maximum action step limit for each task to assess the success rate. If the task was not completed within this limit, it was considered a failure. To set a reasonable threshold, we conducted warm-up tests on 20 samples to estimate the average number of steps required to complete the task. The maximum action step limit was set to 2 times this average value. The success criteria for each task, including target conditions, are detailed in Tab. 6.

We provide a detailed illustration of the overall evaluation pipeline of **RoboMonster** (see Fig. 5). The system first extracts information from the image, and then, through planning in **RoboMonster** Brain, selects suitable end-effectors for the agents $\mathcal{A}_1$ and $\mathcal{A}_2$. The selected end-effectors are employed to execute the tasks using either DP or DP3.

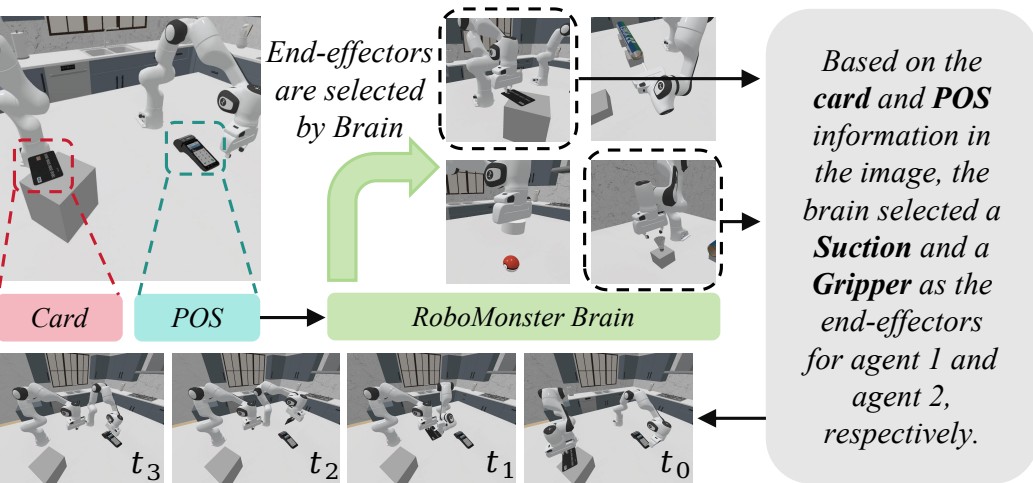

Figure 5: Flowchart of the complete evaluation process, including both planning and execution.

## C    DETAILS OF ROBOMONSTER BRAIN

### C.1    IMPLEMENTATION DETAILS

**Single-Agent Pipeline.** We implement a single-agent pipeline. Images are provided as base64 data URLs when available. End-effector options are strictly derived from the provided robots of the current sample.

**Multi-Agent System.** We also define a conceptual MAS consisting of Analyzer, Planner, Selector, and Validator. Each agent consumes the task context and passes structured intermediate outputs to the next stage. The Validator enforces format and option constraints and triggers retries when confidence is low.

For Single-Agent Pipeline and Multi-Agent System, our relevant call site is:

```
@backoff.on_exception(backoff.expo, Exception, max_tries=5, max_time=60)
def api_call_with_retry(messages, model_name):
    return client.chat.completions.create(
        model=model_name,
        messages=messages,
        temperature=0,
        max_tokens=2000,
```

```
    )
```

## C.2 PROMPT DESIGN

### C.2.1 END-EFFECTOR AVAILABILITY

We construct heterogeneous embodied agent options.

```
    robot_to_end_effector_desc = {
        'stompy': 'Claw hand on Stompy: Stompy is a bipedal robot
    designed for dynamic walking and stomping tasks, featuring
    articulated arms. Color: Light blue body with yellow and orange
    accents. Equipped with a claw hand for grasping objects.',

        'fetch': 'Gripper on Fetch: Fetch is a wheeled robot with a
    flexible arm for object manipulation, designed for mobility and
    dexterity. Color: White with blue and black accents. Uses a gripper
    end-effector: two symmetric \'fingers/plates\' that open and close in
     parallel along the same line. From the front, it looks like two
    parallel small flat plates with a gap in the middle; from the side,
    you can see the \'top/bottom or left/right clamping\' shape.
    Versatile and precise, suitable for most regular or rigid objects
    that can be gripped; adapts to some size variation. May be unstable
    for very thin, slippery, or objects with insufficient gripping
    surfaces.',

        'unitree_h1': 'Dexterous hands on Unitree_H1: Unitree_H1 is a
    humanoid robot with arms and legs designed for human-like movements
    and tasks. Color: Black. Equipped with dexterous hands for complex
    manipulation tasks requiring fine motor control. Best for delicate
    operations and complex assembly tasks. Excellent for precise
    manipulation of various objects.',

        'panda': 'Gripper on Panda: Panda is a fixed robotic arm designed
     for precise and delicate manipulation tasks. Color: White with black
     accents. Uses a gripper end-effector: two symmetric \'fingers/plates
    \' that open and close in parallel along the same line. From the
    front, it looks like two parallel small flat plates with a gap in the
     middle; from the side, you can see the \'top/bottom or left/right
    clamping\' shape. Versatile and precise, suitable for most regular or
     rigid objects that can be gripped; adapts to some size variation.
    May be unstable for very thin, slippery, or objects with insufficient
     gripping surfaces.',

        'unitree_go2': 'Claw hand on Unitree_Go2: Unitree_Go2 is a
    compact quadrupedal robot optimized for agile movement and stability
    with four legs for efficient locomotion. Color: White. Equipped with
    a claw hand for grasping objects.',

        'anymal_c': 'Claw hand on Anymal_C: Anymal_C is a quadrupedal
    robot built for navigating rough terrains and performing complex
    tasks with four articulated legs. Color: Red and black with some
    accents. Equipped with a claw hand for grasping objects.',

        'Suction': 'The end looks like a small round contact pad (not two
     clearly separated jaws). In simulation, it is always closed, without
     a real vacuum mechanism; it \'simulates suction\' by pressing the
    small round pad (actually a closed gripper) normally against the
    object\'s surface. Suitable for smooth, relatively flat targets (such
     as thin cards or flat blocks on a table), most stable when a good
    sealing surface is available. Not suitable for porous/rough/high
    curvature or overly heavy objects, or when there are insufficient
    suction points.',
```

```
    'Circle': 'A short cylindrical tube with an opening. No obvious
jaws or rod-like actuators. The opening cannot be seen from top or
side views, only from below. SPECIFICALLY DESIGNED for round/
spherical objects of all sizes including balls. Provides stable
constraint by \'caging\' round targets like balls, spheres, vases or
buckets that are hard to grip with gripper jaws and not suitable for
suction. IDEAL CHOICE for any ball-related tasks.',

    'Stick': 'A thin, smooth rod with no jaws or suction pad at the
end. Used for pushing, inserting, or clearing in narrow cavities/
channels (e.g., pushing a shuttlecock into a bucket, clearing a thin
pipe); not suitable for carrying or holding objects.'
 }
```

### C.2.2 SYSTEM INSTRUCTION

The system message enumerates the available end-effectors and hard-constrains the answer space:

```
    instruction_following = (
        r'Available end-effectors in this scenario: {robot_set} '
        r'Available end-effector options: {available_end_effectors} '
        r'CRITICAL: You can ONLY choose from the end-effector options
    listed above! These are the ONLY available options for this specific
    scenario. '
        r'Task: Analyze the given task and select the appropriate end-
    effector(s) in the correct execution order. '
        r'IMPORTANT RULES: '
        r'1. You MUST select ONLY from the available end-effector options
     listed above - no exceptions! '
        r'2. If an end-effector is not in the available options list, you
     CANNOT use it, even if it might seem suitable for the task. '
        r'3. For single-step tasks, choose one end-effector. For multi-
    step tasks, list multiple end-effectors in execution order. '
        r'4. Consider object properties (size, shape, weight, material)
    when selecting from the available options. '
        r'5. Consider manipulation requirements (precision, force,
    dexterity) when choosing from available options. '
        r'6. Use the exact end-effector names as shown in the available
    options list. '
        r'Reasoning process: Think through the task step-by-step,
    considering object properties, manipulation requirements, and
    execution sequence, while staying within the constraints of available
     options. '
        r'Response format: Provide your reasoning in <think> </think>
    tags, followed by your final answer in <answer> </answer> tags. '
        r'The answer must be a Python list of end-effector names in
    execution order, using the EXACT names from the available options. '
        r'Example formats: ["claw hand on stompy"] or ["gripper on fetch",
     "dexterous hands on unitree_h1"] for multi-step tasks.'
     )
```

### C.2.3 USER MESSAGE

The user content contains an optional image (base64 data URL) and a text line `"Task Description:  ...".`

**MAS Prompt.** Our Multi-Agent System employs a sequential four-stage pipeline where each agent has specialized prompts:

*Analyzer Agent.* Extracts task components using pattern-based fallback methods:

```
    system_prompt = """You are a task analysis expert specializing in
    robotic manipulation tasks.
```

```
Your role is to analyze the given task description and identify:
1. What objects are involved in the task
2. What actions need to be performed
3. Important properties of the objects (shape, surface, size)

Keep your response simple and clear. Focus on the key information
needed for tool selection."""
```

*Planner Agent.* Creates execution plans with tool requirement analysis:

```
system_prompt = """You are an execution planning expert for robotic
manipulation tasks.

Based on the task analysis, create a detailed execution plan.

PLANNING GUIDELINES:
1. TASK COMPLEXITY ANALYSIS:
- Single-step tasks: One main action (e.g., "move spoon")
- Multi-step tasks: Multiple distinct actions (e.g., "insert bread,
activate toaster, place bowl")

2. STEP IDENTIFICATION:
- Break down the task into atomic actions
- Identify if different actions require different tools
- Note sequence dependencies (what must happen first)

3. TOOL REQUIREMENTS:
- Consider if each step needs a different end-effector
- Flag when precision vs. power tools are needed
- Identify if specialized tools are required (e.g., 'circle' for
balls, 'suction' for cards)

4. OUTPUT FORMAT:
- Clearly state if this is a SINGLE-STEP or MULTI-STEP task
- List each step with its tool requirements
- Highlight any special considerations

Focus on step-by-step breakdown and tool requirement analysis."""
```

*Selector Agent.* Makes final tool selections with enhanced decision logic:

```
system_prompt = """Available end-effectors:

{ROBOT_SET}

CRITICAL TASK ANALYSIS FRAMEWORK:

1. PRECISE TASK TYPE CLASSIFICATION:
a) Simple single-action tasks: "place apple on table", "pick up
banana"
→ Use EXACTLY ONE tool best suited for the primary object

b) Multi-action same-capability tasks: "place banana and apple in
bowl"
→ Use ONE versatile tool that can handle all objects

c) Multi-action different-capability tasks: "insert bread, activate
toaster, place bowl"
→ Use MULTIPLE tools: each action needs different capabilities
→ Bread insertion: delicate manipulation (dexterous hands)
→ Toaster activation: button/lever pressing (gripper/hands)
→ Bowl placement: general manipulation (gripper/hands)

2. ENHANCED OBJECT-TOOL MATCHING RULES:
a) OBJECT-SPECIFIC tools (use ONLY when object demands it):
```

```
1080        - Spherical objects (balls, oranges): 'circle' IF no other tool can
1081        handle safely
1082        - Flat rigid items (cards, plates): 'suction' IF precision placement
1083        needed
1084        - Delicate/soft items (bread, fruit): 'dexterous hands' IF crushing
1085        is a risk
1086
1087        b) VERSATILE tools (prefer when possible):
1088        - 'gripper on fetch': Mobile platform, good for most pick/place tasks
1089        - 'dexterous hands on unitree_h1': Complex manipulation, fine motor
            control
1090        - 'claw hand on stompy/anymal_c': Power tasks, robust grasping
1091
1092        3. TOOL SELECTION DECISION TREE:
1093        Step 1: Count distinct action types needed
1094        Step 2: For each action, determine minimum capability required
1095        Step 3: Find tools that can handle each capability
            Step 4: Minimize tool count while meeting all requirements
1096
1097        Example: "Insert bread, activate toaster, place bowl"
1098        - Action 1: Insert bread → needs delicate manipulation → dexterous
            hands
1099        - Action 2: Activate toaster → needs precise button press → gripper
1100        or hands
1101        - Action 3: Place bowl → needs general manipulation → gripper or
            hands
1102        - Decision: Since actions 2&3 need similar capability but action 1 is
1103         unique
1104        - Result: ['dexterous hands on unitree_h1', 'gripper on fetch']
1105
1106        4. COMMON ERRORS TO AVOID:
1107        - DON'T use multiple tools when one versatile tool can handle
            everything
1108        - DON'T use single tool when actions need genuinely different
            capabilities
1109        - DON'T default to 'gripper on fetch' without considering object
1110        properties
1111        - DON'T use specialized tools (circle, suction) unless truly
1112        necessary
1113
1114        5. PLATFORM CONSIDERATIONS:
1115        - Mobile platforms (fetch, stompy, anymal_c, unitree_h1): Kitchen
            navigation
1116        - Fixed platform (panda): Limited workspace, high precision
1117        - Choose platform based on workspace requirements, not just tool type
1118
1119        VALIDATION CHECKLIST BEFORE SELECTION:
1120        - Have I identified all distinct actions required?
1121        - Does each action need different tool capabilities?
1122        - Can a single versatile tool handle all requirements safely?
1123        - Are specialized tools only used when objects demand them?
            - Is the total tool count justified by genuine capability differences
1124        ?
1125
1126        IMPORTANT: You must provide your final selection in the following
            format:
1127        <answer>['tool1', 'tool2']</answer>
1128        Where 'tool1', 'tool2' are the exact names of the selected end-
            effectors.
1129        For single tool selection, use: <answer>['tool1']</answer>
1130
1131        CRITICAL: For this task, you can ONLY choose from the following end-
1132        effectors: {allowed}. Do NOT select any other tools, even if they
1133        seem suitable."""
```

**Validator Agent.** Provides quality control with retry mechanism:

```
system_prompt = """Available end-effectors:

{ROBOT_SET}

ENHANCED VALIDATION CRITERIA:

1. TOOL COUNT APPROPRIATENESS:
a) Single-action tasks: Should use EXACTLY ONE tool
- "place apple" → ONE tool only
- RED FLAG: Multiple tools for simple placement

b) Multi-action same-capability tasks: Should use ONE versatile tool
- "place banana and apple" → ONE versatile tool
- RED FLAG: Multiple tools when one can handle all objects

c) Multi-action different-capability tasks: Should use MULTIPLE tools
- "insert bread, activate toaster, place bowl" → TWO tools minimum
- Bread needs delicate manipulation, toaster needs button press
- RED FLAG: Single tool for genuinely different capabilities

2. OBJECT-TOOL MATCHING VALIDATION:
a) Specialized tool usage CHECK:
- 'circle' used ONLY for spherical objects that require it
- 'suction' used ONLY for flat items requiring precision
- 'dexterous hands' used for delicate/complex manipulation
- RED FLAG: Specialized tools used unnecessarily

b) Versatile tool preference CHECK:
- When objects can be handled by general tools, prefer them
- 'gripper on fetch' for mobile pick/place tasks
- RED FLAG: Over-specialization when not needed

3. CAPABILITY-REQUIREMENT MATCHING:
a) Action analysis:
- Delicate manipulation → dexterous hands required
- Button/lever activation → precise gripper or hands
- General pick/place → any gripper suitable
- Heavy lifting → robust claw hands

b) Platform requirements:
- Kitchen navigation → mobile platforms (fetch, stompy, unitree_h1)
- Fixed workspace → panda acceptable
- RED FLAG: Platform mismatch with workspace needs

4. CRITICAL ERROR PATTERNS TO FLAG:
a) Tool quantity errors:
- Multiple tools for single-capability tasks (OVERUSE)
- Single tool for multi-capability tasks (UNDERUSE)

b) Object mismatch errors:
- Hard grippers for soft objects when gentle options available
- Specialized tools when general tools suffice
- Missing specialized tools when objects demand them

c) Platform selection errors:
- Fixed platforms for tasks requiring mobility
- Wrong capability level for task complexity

5. SCORING GUIDELINES (STRICTER CRITERIA):
- 0.9-1.0: Perfect tool count + perfect object matching + optimal
platform
- 0.7-0.8: Correct tool count + good matching + appropriate platform
- 0.5-0.6: Minor count/matching issues + acceptable platform choice
```

```
    - 0.3-0.4: Major count errors OR poor matching + suboptimal platform
    - 0.0-0.2: Wrong tool count AND poor matching AND inappropriate
    platform

    6. VALIDATION DECISION TREE:
    Step 1: Count distinct capabilities needed → Expected tool count
    Step 2: Check if each selected tool matches required capability
    Step 3: Verify no over-specialization or under-specialization
    Step 4: Confirm platform choice matches workspace requirements
    Step 5: Score based on how well selection meets all criteria

    COMMON FAILURE PATTERNS TO DETECT:
    - Using 2+ tools when 1 versatile tool can handle everything
    - Using 1 tool when actions genuinely need different capabilities
    - Choosing specialized tools without clear object-specific need
    - Missing mobile platform for kitchen navigation tasks

    Format: <answer>['tool1', 'tool2']</answer>
    Where 'tool1', 'tool2' are the exact names of the selected end-
    effectors.

    CRITICAL: For this task, you can ONLY choose from the following end-
    effectors: {allowed}. Do NOT select any other tools, even if they
    seem suitable.

    After your reasoning, please rate the appropriateness of your
    selected end-effectors for this specific task on a scale from 0.0 (
    completely unreasonable) to 1.0 (perfectly reasonable). Return your
    score in the format <score>0.85</score>.
    """
```

INTER-AGENT COMMUNICATION   Agents pass structured context through a `TaskContext` dataclass containing analysis results, execution steps, tool selections, and validation feedback. The system supports automatic retry when validation confidence falls below 0.6.

## C.3   SCORING AND VALIDATION

We use a task-specific metric `viki_1` that combines accuracy and format compliance:

- **Base Metric**: `score = 0.9 * acc_reward + 0.1 * format_reward`
- **Smart Matching**: We use `smart_compute_score`, which first evaluates `viki_1`. If not perfect, it parses the `<answer>` list and applies a normalization (e.g., mapping `"gripper on panda"` and `"gripper on fetch"` to `"gripper"`) before re-checking sequence equality.

Relevant implementation excerpts:

```
original_score = viki_1.compute_score(predict_str, ground_truth_str)
# If not perfect, normalize and compare element-wise in order
if all(normalize(pred) == normalize(gt) for pred, gt in zip(pred_list,
 gt_list)):
format_score = viki_1.format_reward(predict_str)
return 0.9 * 1.0 + 0.1 * format_score
```

## C.4   MODEL LIST

Tested models include:

- `gpt-5` accuracy: 0.4400
- `gemini-2.5-pro` accuracy: 0.4250
- `Qwen/Qwen2.5-VL-32B-Instruct` accuracy: 0.2350

- `Pro/Qwen/Qwen2.5-VL-7B-Instruct` accuracy: 0.2900
- `claude-sonnet-4-20250514` accuracy: 0.4150
- `glm-4.5v` accuracy: 0.2600

### C.5 DATASET SCHEMA

Each sample provides the following fields used by our runner:

- `prompt`: an array of chat messages; index 1 contains the user message text (prefixed by "Task Description:")
- `images`: optional list with binary bytes; the first item is written to a temporary PNG file for base64 encoding
- `robot`: list of available robots for this scenario; only these determine the end-effector options
- `reward_model.ground_truth`: a stringified Python list (e.g., `"['gripper on panda']"`) representing the reference sequence

The runner maps robots to end-effector option names and builds the system message accordingly. Independent tools (`suction`, `circle`, `stick`) are only included if explicitly present in `robot`.

## D HETEROGENEOUS EFFECTOR AGENTS

In this section, we provide a detailed report on the code implementation and other technical details of heterogeneous multi-end effector embodied agents in the ManiSkill simulator Tao et al. (2024a).

The specific implementation of heterogeneous multi–end effector agents in the simulator is as follows:

1. *Gripper:* We directly load the `panda.urdf` file integrated into ManiSkill.
2. *Suction:* Since accurately modelling the pressure of a suction cup in simulation remains challenging, we maintain the `panda` model with its right and left fingers permanently closed. Upon invocation of the `open_suction()` interface, if a finger is detected in contact with another object, the contacted object is switched from a dynamic to a kinematic state and is constrained to follow the finger. When the `close_suction()` interface is called, the object's dynamic properties are restored.
3. *Stick:* We directly load the `panda_stick.urdf` file integrated into ManiSkill.
4. *Ring-shaped Gripper:* First, we designed `panda_circle.urdf`, adapted for ManiSkill, as the URDF model of a ring-shaped gripper. At the same time, the ring-shaped gripper's wrapper inherits from the suction wrapper: the interface methods are replaced by `open_circle()` and `close_circle()`, while the rest of the logic remains unchanged.

### D.1 SIMULATOR AGENT WRAPPER

Each sim step, if any finger link touches a valid dynamic rigid body, this function performs a one-shot "attach." It flips the target to kinematic, stores the parent-to-target relative pose via `self._relative_pose = fcomp.pose.inv() * tcomp.pose`, and records internal state; returns `True` on success, `False` otherwise.

```python
def grab_once(self) -> bool:
    """Attach a valid dynamic target if any finger link is in contact."""
    if self.is_attached():
        return True
    if not self._finger_links:
        return False

    finger_names = {self._pretty_name(l) for l in self._finger_links}
    for c in self.scene.get_contacts(): # contacts available after each
        ↪ simulation step
```

```
1296        side0, side1 = self._collect_side(c, 0), self._collect_side(c, 1)
1297        for s_touch, s_other in ((side0, side1), (side1, side0)):
1298            finger_entity = self._pick_by_name(s_touch, finger_names)
1299            if not finger_entity:
1300                continue
1301            target_entity = self._pick_valid_target(s_other) # dynamic, not
                    ↪ an articulation link
1302            if not target_entity:
1303                continue
1304
1305            fcomp = finger_entity.find_component_by_type(physx.
                    ↪ PhysxArticulationLinkComponent)
1306            tcomp = target_entity.find_component_by_type(physx.
1307                ↪ PhysxRigidDynamicComponent)
1308            if not (fcomp and tcomp):
1309                continue
1310
1311            # Switch to kinematic and record relative pose (parent^-1 *
                    ↪ target)
1312            tcomp.set_kinematic(True)
1313            self._attached_comp = tcomp
1314            self._parent_link_comp = fcomp
1315            self._relative_pose = fcomp.pose.inv() * tcomp.pose
1316            return True
1317    return False
```

Call every step while attached. It drives the target using the stored relation `target_world = parent_world * relative_pose`; writes `new_pose` to the kinematic body.

```
1321 def update_attachment(self):
1322     """Force␣target␣pose␣=␣parent␣pose␣*␣relative␣pose␣(per-step␣follow)."
            ↪ ""
1323     if not self.is_attached():
1324         return
1325     new_pose = self._parent_link_comp.pose * self._relative_pose
1326     self._attached_comp.entity.set_pose(new_pose) # kinematic body is
1327         ↪ driven externally
```

Detach. Switch the target back to dynamic, zero its linear/angular velocities to prevent bursts, and clear internal state

```
1332 def release(self):
1333     """Restore␣dynamic␣state␣and␣zero␣velocities␣to␣avoid␣bursts␣on␣
            ↪ release."""
1334     if not self.is_attached():
1335         return
1336     try:
1337         self._attached_comp.set_kinematic(False) # back to dynamic
1338         self._attached_comp.linear_velocity = np.zeros(3)
1339         self._attached_comp.angular_velocity = np.zeros(3)
1340     finally:
1341         self._attached_comp = None
1342         self._parent_link_comp = None
1343         self._relative_pose = None
```

## D.2    SIMULATOR RECORD WRAPPER

On each env step, coerce per-agent action vectors to fixed target dims (stick = 7, circle/suction/gripper = 8) by truncating/padding, then forward the normalized action to the base recorder; this keeps dataset shapes consistent without changing environment execution.

```
def step(self, action):
```

```
1350        """Normalize per-agent action dims for recording without altering env
1351            ↪ behavior: - stick: 7D - circle/suction/gripper: 8D"""
1352        dim_map = self._target_action_dim_map()
1353
1354        def _fix_vec(vec: np.ndarray, want: int) -> np.ndarray:
1355            v = np.asarray(vec).reshape(-1).astype(np.float32)
1356            if v.size == want:
1357                return v
1358            if v.size > want:
1359                return v[:want]
1360            out = np.zeros((want,), dtype=np.float32)
1361            out[: v.size] = v
1362            return out
1363
1364        if isinstance(self.env.action_space, gym_utils.spaces.Dict):
1365            # Multi-agent dict: normalize each agent vector to its target dim
1366            assert isinstance(action, dict), "Expect dict action for multi-
1367                ↪ agent env"
1368            norm = {}
1369            keys = (list(self.env.action_space.spaces.keys())
1370                    if hasattr(self.env.action_space, "spaces")
1371                    else list(action.keys()))
1372            for k in keys:
1373                want = dim_map.get(k, int(np.prod(np.asarray(action[k]).shape)))
1374                norm[k] = _fix_vec(action[k], want)
1375        else:
1376            # Single agent
1377            want = dim_map.get("__single__", int(np.prod(np.asarray(action).
1378                ↪ shape)))
1379            norm = _fix_vec(action, want)
1380
1381        # Delegate to base recorder (records + steps env)
1382        return super().step(norm)
```

When flushing, write a complete episode slice into an HDF5 group `"traj_N"`: skip the initial dummy frame, dump observations (gzip for `rgb`/`depth`/`seg`), write per-agent actions (with optional key renaming), flags, states, and rewards; also append JSON episode metadata. Inputs: internal trajectory buffer; output: files on disk plus updated episode index.

```
1384    def flush_trajectory(self,
1385                verbose=False,
1386                ignore_empty_transition=True,
1387                env_idxs_to_flush=None,
1388                save: bool = True):
1389        """Flush a trajectory slice to disk as HDF5 + JSON metadata. Skips
1390            ↪ first dummy frame; compresses images; preserves fixed action
1391            ↪ dims."""
1392        flush_count = 0
1393        if env_idxs_to_flush is None:
1394            env_idxs_to_flush = np.arange(0, self.num_envs)
1395
1396        for env_idx in env_idxs_to_flush:
1397            start_ptr = self._trajectory_buffer.env_episode_ptr[env_idx]
1398            end_ptr = len(self._trajectory_buffer.done)
1399            if ignore_empty_transition and end_ptr - start_ptr <= 1:
1400                continue
1401            flush_count += 1
1402
1403            if save:
                # Create /traj_N
                self._episode_id += 1
                traj_id = f"traj_{self._episode_id}"
                group = self._h5_file.create_group(traj_id, track_order=True)
```

```
1404            # Minimal recursive writer (dicts + arrays), with special
1405              ↪ handling for vision & actions
1406         def recursive_add_to_h5py(group: h5py.Group, data: Union[dict,
1407            ↪ Array], key):
1408           if isinstance(data, dict):
1409               subgrp = group.create_group(key, track_order=True)
1410               for k in data.keys():
1411                   recursive_add_to_h5py(subgrp, data[k], k)
1412           else:
1413               if key in ("rgb", "depth", "seg"):
1414                   group.create_dataset(
1415                       key,
1416                       data=data[start_ptr:end_ptr, env_idx],
1417                       dtype=data.dtype,
1418                       compression="gzip",
1419                       compression_opts=5,
1420                   )
1421               elif group.name.endswith("/actions"):
1422                   # actions already sliced to (T, D) per-agent below
1423                   group.create_dataset(key, data=data[start_ptr+1:end_ptr
1424                      ↪ ], dtype=data.dtype)
1425               else:
1426                   group.create_dataset(
1427                       key,
1428                       data=data[start_ptr:end_ptr, env_idx],
1429                       dtype=data.dtype,
1430                   )

         # Observations
         if self.record_observation:
           obs_buf = self._trajectory_buffer.observation
           if isinstance(obs_buf, dict):
               recursive_add_to_h5py(group, obs_buf, "obs")
           elif isinstance(obs_buf, np.ndarray):
               if self.cpu_wrapped_env:
                   group.create_dataset("obs",
                                   data=obs_buf[start_ptr:end_ptr],
                                   dtype=obs_buf.dtype)
               else:
                   group.create_dataset("obs",
                                   data=obs_buf[start_ptr:end_ptr, env_idx
                                       ↪ ],
                                   dtype=obs_buf.dtype)
           else:
               raise NotImplementedError(f"Unsupported␣obs␣type:␣{type(
                   ↪ obs_buf)}")

         # Episode metadata (JSON sidecar is updated later)
         episode_info = dict(
           episode_id=self._episode_id,
           episode_seed=self.base_env._episode_seed[env_idx],
           control_mode=self.base_env.control_mode,
           elapsed_steps=end_ptr - start_ptr - 1,
         )
         if self.num_envs == 1:
           episode_info.update(reset_kwargs=self.last_reset_kwargs)
         else:
           episode_info.update(reset_kwargs=dict())

         # Slice runtime tensors (skip first dummy frame)
         actions = common.index_dict_array(
           self._trajectory_buffer.action,
           (slice(start_ptr + 1, end_ptr), env_idx),
         )
```

```
1458            terminated = self._trajectory_buffer.terminated[start_ptr+1:
1459               ↪ end_ptr, env_idx]
1460            truncated = self._trajectory_buffer.truncated [start_ptr+1:
1461               ↪ end_ptr, env_idx]
1462
1463            # Optional agent key normalization (example: rename stick agent
                  ↪ key)
1464            def _rename_agent_key_for_write(k: str) -> str:
1465                if isinstance(k, str) and k.startswith("panda_stick"):
1466                    return "panda" if "-" not in k else "panda-" + k.split("-",
1467                       ↪ 1)[1]
1468                return k
1469
1470            # Actions
1470            if isinstance(self._trajectory_buffer.action, dict):
1471                actions_to_write = {}
1472                for k, v in actions.items():
1473                    new_k = _rename_agent_key_for_write(k)
1474                    if (new_k in actions_to_write) and (new_k != k):
1475                        new_k = k
1475                    actions_to_write[new_k] = v
1476                recursive_add_to_h5py(group, actions_to_write, "actions")
1477            else:
1478                group.create_dataset("actions", data=actions, dtype=np.
                      ↪ float32)
1479
1480            # Flags
1481            group.create_dataset("terminated", data=terminated, dtype=bool)
1482            group.create_dataset("truncated", data=truncated, dtype=bool)
1483
1484            # Success / fail (optional)
1484            if self._trajectory_buffer.success is not None:
1485                end_ptr2 = len(self._trajectory_buffer.success)
1486                group.create_dataset(
1487                    "success",
1488                    data=self._trajectory_buffer.success[start_ptr+1:end_ptr2,
                          ↪ env_idx],
1489                    dtype=bool,
1490                )
1491                episode_info.update(success=self._trajectory_buffer.success[
                      ↪ end_ptr2 - 1, env_idx])
1492
1493            if self._trajectory_buffer.fail is not None:
1494                group.create_dataset(
1495                    "fail",
1495                    data=self._trajectory_buffer.fail[start_ptr+1:end_ptr,
                          ↪ env_idx],
1496                    dtype=bool,
1497                )
1498                episode_info.update(fail=self._trajectory_buffer.fail[end_ptr
                      ↪ - 1, env_idx])
1499
1500
1501            # Environment states (if enabled)
1502            if self.record_env_state:
1503                recursive_add_to_h5py(group, self._trajectory_buffer.state, "
                      ↪ env_states")
1504
1505            # Rewards (if enabled)
1506            if self.record_reward:
1507                group.create_dataset(
1508                    "rewards",
1508                    data=self._trajectory_buffer.reward[start_ptr+1:end_ptr,
                          ↪ env_idx],
1509                    dtype=np.float32,
1510                )
1511
```

```
1512             # Update manifest JSON
1513             self._json_data["episodes"].append(episode_info)
1514             dump_json(self._json_path, self._json_data, indent=2)
1515
1516             if verbose:
1517                 print(f"Recorded episode {self._episode_id}")
1518
1519     # Truncate in-memory buffer to save RAM and advance per-env pointers
1520     if flush_count > 0:
1521         self._trajectory_buffer.env_episode_ptr[env_idxs_to_flush] = len(
1522             ↪ self._trajectory_buffer.done) - 1
1523         min_env_ptr = self._trajectory_buffer.env_episode_ptr.min()
1524         N = len(self._trajectory_buffer.done)
1525
1526         if self.record_env_state:
1527             self._trajectory_buffer.state = common.index_dict_array(self.
1528                 ↪ _trajectory_buffer.state, slice(min_env_ptr, N))
1529         self._trajectory_buffer.observation = common.index_dict_array(self.
1530             ↪ _trajectory_buffer.observation, slice(min_env_ptr, N))
1531         self._trajectory_buffer.action = common.index_dict_array(self.
1532             ↪ _trajectory_buffer.action, slice(min_env_ptr, N))
1533         if self.record_reward:
1534             self._trajectory_buffer.reward = common.index_dict_array(self.
1535                 ↪ _trajectory_buffer.reward, slice(min_env_ptr, N))
1536         self._trajectory_buffer.terminated = common.index_dict_array(self.
1537             ↪ _trajectory_buffer.terminated, slice(min_env_ptr, N))
1538         self._trajectory_buffer.truncated = common.index_dict_array(self.
1539             ↪ _trajectory_buffer.truncated, slice(min_env_ptr, N))
1540         self._trajectory_buffer.done = common.index_dict_array(self.
1541             ↪ _trajectory_buffer.done, slice(min_env_ptr, N))
1542         if self._trajectory_buffer.success is not None:
1543             self._trajectory_buffer.success = common.index_dict_array(self.
1544                 ↪ _trajectory_buffer.success, slice(min_env_ptr, N))
1545         if self._trajectory_buffer.fail is not None:
1546             self._trajectory_buffer.fail = common.index_dict_array(self.
1547                 ↪ _trajectory_buffer.fail, slice(min_env_ptr, N))
1548         self._trajectory_buffer.env_episode_ptr -= min_env_ptr
```