# OpenReview forum: "RoboMonster: Compositional Generalization of Heterogeneous Multi-End Effector Embodied Agents"
_ICLR.cc/2026/Conference — ICLR 2026 Conference Withdrawn Submission_

### Official Review · Reviewer_jd4T · 2025-10-21

**Soundness:** 2
**Presentation:** 3
**Contribution:** 2
**Rating:** 4
**Confidence:** 3

**Summary:**

This paper presents RoboMonster, a framework for compositional generalization across heterogeneous embodied agents. Instead of relying on a single morphology, RoboMonster composes multiple robots with diverse end-effectors (e.g., gripper, suction, stick) to handle varied tasks. It employs an MLLM-based Planner–Verifier framework guided by a Robot Manual and evaluates performance on RoboMonster-E, a ManiSkill-based benchmark using imitation learning (DP, DP3). Experiments demonstrate improved performance and compositional generalization over single-agent baselines.

**Strengths:**

* The paper is reasonably well-written and provides ample details on its technical contributions.
* The paper introduces two reusable benchmarks, RoboMonster-P and RoboMonster-E, which can serve as valuable testbeds for future research on heterogeneous embodiment.

**Weaknesses:**

* All evaluations are limited to simulation. It lacks physical robot experiments or robustness analysis under real-world constraints, which limits the practical validation of the proposed approach.
* The overall pipeline shows limited novelty, as numerous prior studies about agent have employed comparable plan–execution frameworks built upon large language models.
* The approach relies heavily on intricate prompt engineering, which is a labor-intensive and somewhat cumbersome process.

**Questions:**

* See weaknesses.
* The experiments are entirely simulation-based. Could the method support sim-to-real transfer, and how might performance be affected under real-world constraints?
* Which MLLM and LLM are used in the MAS system? The paper does not seem to specify this, and the choice of model may significantly affect performance.
* Have the authors considered comparisons with other hierarchical planning approaches to better contextualize the performance of the proposed system?

---

### Official Review · Reviewer_dpES · 2025-10-25

**Soundness:** 2
**Presentation:** 2
**Contribution:** 2
**Rating:** 2
**Confidence:** 4

**Summary:**

This paper proposes RoboMonster, a framework for combining multiple heterogeneous robots or end-effectors to achieve better task performance. The system has two parts: a planning benchmark (RoboMonster-P) where a multimodal LLM selects suitable agents using a predefined “Robot Manual,” and an execution benchmark (RoboMonster-E) built in ManiSkill with four specialized end-effectors. The experiments show that using multiple effectors improves task success compared to using a single gripper, and that the planner–verifier pipeline achieves slightly better planning accuracy than large vision-language models alone.

**Strengths:**

- The paper explores a relevant and increasingly important question: how to make robots work together effectively when they have different physical capabilities. Instead of focusing on a single “universal” morphology, it argues for combining specialized agents for maximum performance.
- The design of a “Robot Manual” that explicitly lists each effector’s skills and limits is a straightforward way to bring affordance reasoning into the LLM’s decision process.
- The separation between high-level planning and low-level control is structured and makes it easy to understand the system’s logic.
- The experimental results show that heterogeneous setups outperform single-effector ones across a range of tasks, and the proposed benchmarks (RoboMonster-P and -E) could be useful for future studies of heterogeneous collaboration.

**Weaknesses:**

- Presentation and clarity of the paper need substantial improvement. Some of the paper’s key terms are abused, e.g., I don't think having heterogeneous end-effectors collaborating on a task is related to "compositional generalization", or there is an "embodiment gap" between simulated and real-world robots in this paper's context.
- While the planner–verifier architecture is reasonable, its contribution feels limited and can't justify the design complexity - the overall accuracy gain over GPT-5 is marginal, and it's not clear how much of the improvement comes from manually specified rules rather than learning.
- The Robot Manual itself is heavily hand-engineered, which makes the system feel more like a rule-based decision pipeline than an emergent reasoning framework. The downside of engineering a manual is reduced flexibility when objects to be manipulated fall outside the rules.
- No real-world robot experiments and results, which can bring additional challenges to perception and manipulation for verifying the proposed approach's competence.
- Finally, the conclusion that “multiple specialized end-effectors work better than one” is intuitive and expected; the paper would benefit from deeper insights on why and when the combination helps most.

**Questions:**

- In a single sentence, what are the key insights and contributions that previous papers fail to show? It's not obvious to me what the entirely new findings and breakthroughs are in this paper.
- What's the specific LLM/VLM used in MAS?

---

### Official Review · Reviewer_Ldcd · 2025-11-01

**Soundness:** 2
**Presentation:** 3
**Contribution:** 3
**Rating:** 6
**Confidence:** 4

**Summary:**

This paper proposes a novel system that combines heterogeneous embodied agents to bridge the gap between simulation-based decision-making and real-world robotic constraints. It employs a two-stage framework: a high-level planning phase with MLLM that selects optimal agents and a low-level phase that executes actions based on the selected agents.

**Strengths:**

* The paper introduces a novel and practical paradigm that leverages heterogeneous embodied agents for suitable tasks.
* The paper designs comprehensive benchmarks (RoboMonster-P for planning, RoboMonster-E for execution) covering diverse tasks.
* Extensive experiments and ablations clearly demonstrate the superiority of heterogeneous agent composition: it outperforms single-agent or single-effector baselines significantly

**Weaknesses:**

* The execution validation of RoboMonster is solely conducted in the ManiSkill simulation environment, lacking evaluation on real-world robots.
* This may be less practical in real-world applications, as it is very difficult for an actual robot to switch an end effector for each task.
* The specific design of end effectors would hinder the generality of the model. Like data collection for each end effector.

**Questions:**

* The paper assumes the optimal agents for the low-level agents in the experiment. What if using the agents selected in the high-level phase?
* The paper collects data for the agent selection, by annotation GPT-4o and human correction. Can GPT-4o work for this specific problem? What's the accuracy without human correction?
* Since MAS has low accuracy for agent selection, is there any solution to select unsuitable agents?

---

### Official Review · Reviewer_UfgJ · 2025-11-06

**Soundness:** 2
**Presentation:** 3
**Contribution:** 1
**Rating:** 2
**Confidence:** 4

**Summary:**

The paper presents RoboMonster as a new approach and system for leveraging heterogeneous embodied agent capabilities in longer-range/multi-step task completion. RoboMonster splits the problem into a scheduling level, which allocates subtasks to different embodied capabilities through a MLLM Planner-Verifier module; and an execution level, where models for 4 embodiments are trained on simulator-collected data and evaluated.

For each level, the paper introduces a new benchmark: RoboMonster-P and RoboMonster-E. Experimental results showcase the performance of the proposed scheduler/allocator and the execution performance of 4 different end-effector setups over 4 manipulation tasks.

**Strengths:**

The paper presents an interesting vision of leveraging heterogeneous embodiments as a multi-agent system for task compeltion to better make use of flexible decision-making capabilities of recent foundation modela and minimize impact of specific embodiment constraints of real physical robots in accomplishing realistic tasks.

RoboMonster proposes a practical hierarchical structure to decouple parts of the problem and facilitate validation. Also the created benchmaks and datasets can be of potential interest to the community.

**Weaknesses:**

While the motivation is very promising, the contributions presented in the paper seem of somewhat limited impact. The paper identifies and targets a real issue, but the presented actual scenarios and evaluations are too simplified.

As one of the core claims in the paper relates to compositionality, having only one arm robot with 4 different manipulators and evaluating it only on 5 tasks doesn't really demonstrate the potential of the approach or that "capability-driven composition is a viable route". Even though I particularly believe it is. Moreover, the large numbers of demonstrations for longer-horizon tasks mentioned in the paper also suggest the tested approach may have issues scaling to more complex cases.

The evaluation presented over RoboMonster-P focuses on allocation validity only. However, the feasibility of plans is also a core criteria that should be taken into account (along with re-planning). Unfortunately, even for allocation, the results showed are lower than expected (MAS only .45 over GPT .44). The evaluation of execution also assumes that the scheduling step made correct assingments, but that doesn't hold according to the MAS results presented.

Critically, there is no detailed description of the data/benchmark. No statistics, samples, etc. We never see what actually the plans or tasks decompositions are. Such details are essential to fully evaluate the contribution of the new dataset/benchmark.

Lastly, the concept itself, claimed as first contribution, is not entirely novel, as multi heterogeneous agents system exist for a while, and the specifics of the experimental setup presented is more akin to tool use than really multi-agency composition or collaboration.

**Questions:**

The paper claims experiments used a 200-sample test set from RoboMonster-P, with details in Appendix C. Did I miss the details somehow?

What are the licenses under which RoboMonster-P and RoboMonster-E would be released? This is crucial information to estimate impact of the new benchmark. Would the collected data or models be released too? Under which license?

Just to confirm for clarity... By "global view" you mean global scene view from a fixed camera, correct?

Figure 1 does not really describe the motivation and proposed approach well. I'd suggest replacing it to better reflect the setting.

Minor:
- Fix citation for OpenVLA in intro. Missing year. There are a couple other similar cases in the text.
- "MAS planning" is used in intro without defining Multi-Agent Systems first.

---

### Note · Authors · 2025-11-14

I have read and agree with the venue's withdrawal policy on behalf of myself and my co-authors.